# Determination of the Amino Acid Recruitment Order in Early Life by Genome-Wide Analysis of Amino Acid Usage Bias

**DOI:** 10.3390/biom12020171

**Published:** 2022-01-21

**Authors:** Mingxiao Zhao, Ruofan Ding, Yan Liu, Zhiliang Ji, Yufen Zhao

**Affiliations:** 1Department of Chemical Biology, Key Laboratory for Chemical Biology of Fujian Province, College of Chemistry and Chemical Engineering, Xiamen University, Xiamen 361005, China; 20520180155152@stu.xmu.edu.cn; 2State Key Laboratory of Cellular Stress Biology, School of Life Science, Xiamen University, Xiamen 361102, China; 21620170155349@stu.xmu.edu.cn; 3Institute of Drug Discovery Technology, Ningbo University, Ningbo 315221, China

**Keywords:** LUCA, pseudotime analysis, genetic codon, proteogenesis, recruitment

## Abstract

The mechanisms shaping the amino acids recruitment pattern into the proteins in the early life history presently remains a huge mystery. In this study, we conducted genome-wide analyses of amino acids usage and genetic codons structure in 7270 species across three domains of life. The carried-out analyses evidenced ubiquitous usage bias of amino acids that were likely independent from codon usage bias. Taking advantage of codon usage bias, we performed pseudotime analysis to re-determine the chronological order of the species emergence, which inspired a new species relationship by tracing the imprint of codon usage evolution. Furthermore, the multidimensional data integration showed that the amino acids A, D, E, G, L, P, R, S, T and V might be the first recruited into the last universal common ancestry (LUCA) proteins. The data analysis also indicated that the remaining amino acids most probably were gradually incorporated into proteogenesis process in the course of two long-timescale parallel evolutionary routes: I→F→Y→C→M→W and K→N→Q→H. This study provides new insight into the origin of life, particularly in terms of the basic protein composition of early life. Our work provides crucial information that will help in a further understanding of protein structure and function in relation to their evolutionary history.

## 1. Introduction

Universal common ancestry (UCA) is a core pillar of modern evolutionary theory [1], which was first proposed by Darwin in *On the Origin of Species*. UCA theory is grounded in the hypothesis that all organisms are the offspring of an ancient single life form so-called the last universal common ancestor (LUCA) [2,3]. A further hypothesis proposes that LUCA may be a group of genetically differentiated organisms that inhabited specific ancient environments [4]. In turns, Woese also proposed that LUCA could have been a diverse cell community that survived and evolved as a biological unit based on the existing genetic annealing model of life [5]. Both studies suggested that the contemporary system of the genetic code in living organisms may be different from that existing in LUCA. In the book *Genesis and Evolutionary Development of Life*, Oparin proposed that prokaryotes may have emerged three to four billion years ago with a highly ambiguous and/or primitive genetic sequence encoding proteins built from about seven most abundant amino acids in the primordial soup. Therefore, LUCA may just have had a simpler genetic codon system than that in present-day life forms [6], which gained higher complexity along the process of life evolution [7,8].

It is considered that the process of amino acid acquisition and reduction during life evolution follows the pattern assumed in the neutral model theory [9,10]. Specifically, newly recruited amino acids will sacrifice the use of older ones to gain a foothold [7,11,12]. However, the introduction of new amino acids must be slow because frequent changes in protein composition can be fatal to life [13,14]. As a consequence, amino acids that were recruited early likely have more frequent usage than those recruited later [13,15,16]. In 1975, Wong proposed an evolutionary map of the genetic code and defined two minor amino acid centres (Phe-Tyr and Val-Leu) [17]. In 2000 and 2004, Trifonov suggested the temporal appearance order of amino acids and their respective codons based on various criteria such as thermostability, complementarity, and processivity [7,18]. The coevolution of amino acids and genetic codons is still ongoing today [19,20] and is supported by several recent findings. For example, in the mitochondria of hemichordates, the AAA codon (encoding lysine) was reassigned to asparagine [21]. The AGA/AGG codon (which encodes arginine) was reassigned to serine in the mitochondrial genome of most invertebrates or to glycine in tunicate mitochondria [22]. The UGA codon (stop codon) was reassigned to tryptophan in *Mycoplasma* spp. [23] or to cysteine in *Euplotes octocarinatus* [24]. Most excitingly, recent research has discovered the correspondence between the UGA and UAG codons and new proteinogenic amino acids, i.e., selenocysteine (U) [25] and pyrrolysine (O) [26], respectively. In contrast to advances in the discovery of new codon usage and new amino acids, limited work has been conducted on the recruitment order of amino acids in early life forms.

Life has continued to evolve since its emergence. The imprint of its evolution has left traces recorded in the genome [27]. In this study, we investigated the controversial issue of how amino acids were recruited into LUCA and evolved to their current status in present-day species. For this purpose, we conducted a comparative analysis of amino acid usage and genetic codon bias in large-scale modern organisms. With these pieces of information, we estimated the quasi-time of species emergence. We also reconstructed the recruitment order and routes of 20 amino acids into LUCA and modern species.

## 2. Materials and Methods

### 2.1. Genomic Information

Genomic information was taken from the genome reports of the National Center for Biotechnology Information (NCBI), which were downloaded from ftp.ncbi.nlm.nih.gov/genomes/GENOME_REPORTS/ (as of 19 September 2019). Only the 7043 species with complete genomes in genome reports status column were selected. Viruses were not included in this study. The coding DNA sequence (CDS) and corresponding protein sequence of the selected species were acquired accordingly to GenBank annotations (GCA). The files _cds_from_genomic.fna.gz and _protein.faa.gz for each species were downloaded from ftp://ftp.ncbi.nlm.nih.gov/genomes/all/GCA/, accessed on 19 September 2019. We also collected protein sequences and CDS of 227 eukaryotes from the Ensembl database (release 98). The files .cds.all.fa.gz and .pep.all.fa.gz for the species were downloaded from ftp://ftp.ensembl.org/pub/release-98/fasta/, accessed on 19 September 2019. Overall, 7270 species were included in the study, including 6705 bacterial species, 305 archaeal species and 260 eukaryotes. The distribution of the species across the three domains of life is presented in Appendix A, and the genomic details of the species, including taxonomy, GC%, status and FTP path are shown in Appendix A.

### 2.2. Calculation of Amino Acid Usage and Genetic Codon Usage—Python Scripts

#### 2.2.1. Amino Acid Usage

For each species, genome-wide amino acid usage (excluding uncommon amino acids, such as selenocysteine and pyrrolysine) was calculated as follows:(1)$$F_{i}=\frac{N_{i}}{N_{t}}$$
where *F_i_* is the usage of amino acid *i* in the species, *N_i_* is the count of amino acid *i* and *N_t_* is the total count of twenty amino acids in the species. We calculated theoretical amino acid usage by dividing the number of codons of the corresponding amino acids by 61 (excluding three stop codons) [20] as follows:(2)$$F_{it}=\frac{C_{i}}{61}$$
where *F_it_* is the theoretical usage of amino acid *i* and *C_i_* is the codon number of the corresponding amino acid *i*.

The ranking of amino acid usage across the three domains of life was determined as follows: for every species, the 20 amino acids were ranked from 1 to 20 according to how commonly they were used, with the most common amino acid assigned a value of 20 and the least common one assigned a value of 1. Amino acid usage in each domain of life was thus determined by calculating the sum of the scores of all involved species.

The coefficient of variation (*CV*) is a normalized measure of the dispersion degree of the probability distribution and was calculated as follows.
(3)$$CV=\frac{\sigma_{i}}{\mu_{i}}$$
where $\sigma_{i}$ and $\mu_{i}$ are standard deviation and the mean of amino acid i usage, respectively.

#### 2.2.2. Genetic Codon Usage

Similar to amino acid usage analysis, genetic codon usage, excluding the non-canonical codons (N: any base, purine [R]: A and G bases, pyrimidine [Y]: T and C bases, or H: T, C and A bases) and stop codons, was calculated as follows:(4)$$F_{c}=\frac{N_{c}}{N_{tc}}$$
where *F_c_* is the frequency of codon *c* in all CDS in the species, *N_c_* is the count of codon *c* in the CDS and *N_tc_* is the total count of all 64 codons (including stop codons).

The fold change (*FC*) was introduced to measure the codon usage difference between the two species groups as follows:(5)$$FC=\frac{c_{\geq 45}^{i}}{c_{<45}^{i}}$$

Considering that the GC content of eukaryotic genomes was between 40% and 45%, we set the GC-content threshold to 45%. Thus, FC is the ratio of the average codon usage in species with a GC content ≥45% to those with a GC content of <45%. $c_{\geq 45}^{i}$ and $c_{<45}^{i}$ are the average usage of codon *i* in species with a GC content of ≥45% and <45%, respectively.

### 2.3. Correlation Analysis—Corrplot R Packages

#### 2.3.1. Usage Correlation between Amino Acids or Codons

The overall usage correlation between two amino acids (or codons) *p* and *y* in all species was calculated using the Pearson correlation coefficient:(6)$$P_{p,y}=\frac{cov\left(p,y\right)}{\sigma \left(p\right)\sigma \left(y\right)}$$
where *cov* is the covariance of usage between *p* and *y*, *σ* is the standard deviation for amino acid (or codon) usage of *p* and *y*.

#### 2.3.2. Correlation of Physicochemical Properties between Amino Acids

A total of 566 physicochemical properties of 20 amino acids were derived from the AAindex database (https://www.genome.jp/aaindex/, accessed on 19 September 2019). These properties include multiple aspects of physics, chemistry, protein structure and distribution, transfer-free energy, etc. (Appendix A). The amino acid properties were first normalized using the *z-score*:(7)$$\mathrm{z-score}=\frac{x-\bar{x}}{\sigma }$$
where *x* is the raw value of the properties, *σ* is the standard deviation and is the mean of the properties. The correlation between *X* and *Y* amino acids and physiochemical properties can be calculated using the Spearman correlation coefficient:(8)$$r_{s}=1-\frac{6\sum_{i=1}^{n}d_{i}^{2}}{n(n^{2}-1)}$$
where *r_s_* is the Spearman correlation coefficient, *d_i_* is the ranking difference of *X* or *Y* amino acid properties, and *n* is the total rank, which is 566. 

### 2.4. Phylogenetic Analysis

The dendextend and pvclust R packages (method.dist = ‘correlation’, method.hclust = ‘average’) were used to reconstruct phylogenetic tree, using data on genetic codon and amino acid usage for all the included species. For this purpose, the codon and amino acid usage profiles for each species were represented by the 64-element vector $\chi =\left\{x_{1},\ldots ,x_{n}\right\}$ (*n* = 64) (Appendix A) and the 20-element vector $\chi =\left\{x_{1},\ldots ,x_{n}\right\}$ (*n* = 20) (Appendix A).

### 2.5. Quasi Evolutionary Time Estimation

We applied pseudotime analysis to estimate the quasi-evolutionary time of species based on differences in the genetic codon usage among species. For every species, the codon usage profiles were represented by 64-element vectors $\chi =\left\{x_{1},\ldots ,x_{n}\right\}$ (*n* = 64) (Appendix A). Taking the codon usage vectors as the input, pseudotime analysis was performed using R package monocle 2.20.0 (norm_method = ‘log’, reduction_method = ‘DDRTree’), in which both methods of independent component analysis and minimum spanning tree were implemented [28].

## 3. Results

### 3.1. Genome-Wide Exploration of Amino Acid Usage Bias and Genetic Codon Usage Bias in the Three Domains of Life

#### 3.1.1. Amino Acid Usage Landscape

We provided a portrayal of the genome-wide landscape of amino acid usage across the three domains of life. As presented in Figure 1a, the eukaryotes showed consensus amino acid usage in different phyla. Comparatively, high diversity was shown between different phyla in bacteria and archaea. This may have been caused by the quicker and more differential evolutionary speed of bacterial and archaeal species. The present results directly contrast the theoretical expectations that amino acids encoded by a larger number of codons would have higher usage. Among the three domains of life, only eukaryotes displayed correspondence to this assumption, but there were several exceptions. Arginine (R) has six codons, but its usage in eukaryotes ranked ninth (Figure 1a). Isoleucine (I) has three codons, but it had less usage than aspartic acid (D) and glutamine (Q), which have two codons each. Glutamic acid (E) had two codons but had much higher usage. Lysine (K) also had higher than anticipated usage. A similar trend of amino acid usage was also observed in bacteria and archaea; however, the different phyla of these two domains of life exhibited differential preferences for amino acids. Likewise, CV (coefficient of variation) analysis indicated that the majority of amino acids in Eukaryota had low usage change (CV < 15), except alanine (A) and asparagine (N). In contrast, the CVs of most amino acids in bacteria and the archaea varied (Appendix A). These findings suggest that amino acid usage bias is ubiquitous in the three domains of life. The number of codons is not the sole factor but is a major one for determining amino acid usage in life.

#### 3.1.2. Genetic Codon Usage Landscape

We also identified the genetic codon usage landscape in the three domains of life, upon which the hierarchical clustering of species was performed. Unsurprisingly, the majority of eukaryotic species were prone to clustering together (Figure 1b). In contrast, the archaeal species were interspersed in bacterial species, indicating a similar pattern of codon usage that probably emerged at similar time. In addition, we used fold change (FC) to measure differences in codon usage across different species groups of GC content (Table 1). In most cases, codons ending with G/C had comparatively higher use in species with high GC content than in those with low GC content. Accompanying with the increase of GC content over the three domains of life, the usage of codons ending with G/C increased and the usage of codons ending with A/T decreased, and vice versa (Appendix A). These results support the previous findings that codon usage is correlated with the GC content regardless of the non-coding component of the genome.

We examined the use of synonymous codons (with only terminal bases differing) of the same amino acids. All twenty amino acids, except M and W, are coded by at least two synonymous codons. The synonymous codons for the same amino acid exhibited differential usage across the three domains of life (Figure 1b). For example, the FC values for two synonymous codons (AAT and AAC) of asparagine were 0.29 and 1.23, respectively, indicating that AAT was highly favoured in species with low GC-content, and AAC was preferred in species with high GC-content (Table 1). For amino acids with more than two synonymous codons (i.e., A, G, L, P, R, S, T and V), codon usage bias was more severe than in the case of G/C-ending codons.

Noteworthily, we also examined the effect of any single base variation of codons on the usage in species with different GC content. The data indicate that the codon usage trend was almost reached consensus once the composition of any two bases within codon sequence (for example, AAN, NNA and ANA) was the same (Appendix A); however, when only one base was fixed, the variation caused by the latter codon position was significantly higher than that in the other two positions [29] (Appendix A). This result further supports the contention that the degeneracy of the codon could be an effective rescue mechanism for long-term evolution to countermine the fatal consequences of rapid changes in amino acid usage while tolerating small evolutionary steps.

### 3.2. Chronological Order of Species Emergence Based on Codon Usage Bias

Based on the LUCA hypothesis and other theories of neutral evolution, we assumed that codon usage changed stepwise across long-term evolution to reach current status. The tracking of codon and amino acid usage enabled determination of the evolutionary order of species emergence (Figure 2a,b). The phylogenetic analysis based on codon usage information was more sensitive in depicting early evolutionary relationships than amino acid usage data. The bias between codon and amino acid usage were determined based on the resolved phylogenetic trees for 77 model organisms (Bacteria: 21 species, Archaea: 9 species and Eukaryota: 47 species, Appendix A, Figure 2c). Intriguingly, we found that the phylogenetic relationships were highly similar for codon and amino acid usage. For example, the branches of the advanced eukaryotes and bacterial/archaeal species were highly similar in two phylogenetic trees, and the same species were grouped into small branch leaves (i.e., *Mus musculus* and *Rattus norvegicus*, *Aeropyrum pernix* and *Pyrobaculum aerophilum*, *Escherichia coli* and *Salmonella enterica*). In summary, both codon and amino acid usage mapped the evolutionary relationship of species, but codons allowed better determination of the detailed relationships among species.

Therefore, we estimated the quasi-evolutionary time order of species emergence by conducting pseudotime analysis based on the codon (64 genetic codons) usage profile (Figure 3b). According to the evolutionary order determined in this study, *Actinobacteria* and *Euryarchaeota* were the first prokaryotes and archaeal organisms, respectively, to appear. The first eukaryotic species to emerge was *green algae*, which is consistent with previous findings that *Grypania spiralis* may have been one of the earliest species. According to available fossil data, the mentioned taxon may have appeared approximately 2 billion years ago [30]. The most modern species in all domains of life was the *Betaproteobacteria*, and the most modern species in Archaea and Eukaryota were *Nanoarchaeota* and protists, respectively. Hominids and primates appeared near the middle of the quasi-evolutionary order of eukaryotic species (Figure 3b and Appendix A). In summary, the pseudotime analysis results thus support the hypothesis that Bacteria appeared first, followed by Archaea, and finally Eukaryota as descendants from Archaea. It is noteworthy mentioning that available phylogenetic data indicate that the evolution of codon usage is still ongoing. The short time of inter-generation transition together with the high rate of DNA recombination in bacterial and archaeal species contributed to their more rapid evolution than that of most eukaryotes, except for some single-cell species-such as protists. It remains unclear whether and when the codon usage achieved a fixed ratio.

### 3.3. LUCA Protein Amino Acid Composition and Amino Acid Recruitment Order into Protein

To address the question of amino acid recruitment in early lifeforms, we set several rules (Figure 3a). (1) Rule I: all amino acids were categorised as either old or new. The amino acids detected by Miller’s spark tube experiments and meteorite evidence [31] and those that have fallen out of use in all domains of life during evolution may have been recruited earlier. The old amino acids (A, D, E, G, I, L, P, S, T and V) were easier to synthesise with natural inorganic substances such as ammonia and methane in the primordial environment and have been detected in meteorites; thus, they were most likely to have been recruited in early lifeforms. We assumed that the newly recruited amino acids need time to establish a foothold by sacrificing the usage of the original amino acids [7,11,12]. Accordingly, the use of six amino acids (A, G, P, R, V and W) was largely lost (R^2^ ≳ 0.5), whereas the use of five amino acids (F, I, K, N and Y) was largely gained (R^2^ ≳ 0.5) throughout evolution (Figure 3d). The amino acids whose use was lost were likely to be the ones involved in the proteogenesis of early life. Of these, R was not synthesised in Miller’s experiment; however, it may have been the only basic amino acid to balance the acidity of E and D in LUCA proteins, because R has six codons, while the other two basic amino acids, K and H, have only two codons. Moreover, the usage of the amino acid R (R^2^ = 0.8766) is significantly negatively correlated with evolutionary time. Furthermore, I and L are isomers of each other, and both are, similar to V, branched-chain amino acids. These three amino acids may not all be recruited in the LUCA protein in priority subject to the principle of maximum parsimony. Because I has three codons, L and V have six and four codons, respectively, and the usage of the amino acid I (R^2^ = 0.7881) was significantly positively correlated with evolutionary time, in contrast to amino acids L (R^2^ = 0.1092) and V (R^2^ = 0.4975). The old amino acids A, D, E, G, L, P, R, S, T and V might be first recruited into LUCA proteins; C, F, H, I, K, M, N, Q, W and Y are the new amino acids. (2) Rule II: the order of recruitment of the remaining amino acids into protein. We performed Spearman correlation analysis based on 566 normalised physicochemical properties of 20 amino acids. The amino acids were categorised into two groups according to similarity: the hydrophobicity group (A, C, F, I, L, M, V, W, Y) and the hydrophilicity group (D, E, G, H, K, N, P, Q, S, T, R) (Figure 3c). The correlation coefficients between these two amino acid groups were generally less than 0.5, indicating that they may have been recruited via two independent routes: the hydrophobic and the hydrophilic. It is noteworthy that both groups consist of old and new amino acids. The hydrophobic group contains three old amino acids (A, L and V) and six new ones (C, F, I, M, W and Y); the hydrophilicity group contains seven old amino acids (D, E, G, P, R, S and T) and four new ones (H, K, N and Q). These two groups may have played complementary roles in LUCA proteogenesis. The amino acids recruited earlier would have a comparatively higher usage ranking [13,15,16] and more codons. Taking into consideration the usage rank and current biological synthesis pathways, we speculated that the new amino acids may be recruited into the proteogenesis in two parallel manners: I→F→Y→C→M→W and K→N→Q→H.

### 3.4. Impact of Codon Selection on Amino Acid Usage and Recruitment

To determine the impact of genetic codons on amino acid recruitment, we performed several correlation analyses based on the usage profile of the amino acid or genetic codons across the three domains of life. All 61 genetic codons except TGA, TAG and UAA, were grouped into two clusters (Pearson correlation coefficient r > 0.5) (Figure 4a): one cluster included 29 codons, which all end with G/C, and the other included 28 codons, which all end with A/T. Within each cluster, the codons exhibited significantly positive correlations with each other; however, between the two clusters, the codons were negatively correlated. Codon usage indicates an obvious preference subject to the last base of a given codon. Similar clustering analysis was performed based on amino acid use across the three domains of life, which also indicated two major clusters for 20 amino acids: one cluster consisted of five amino acids (F, I, K, N and Y), and the other cluster consisted of six amino acids (A, G, P, R, V and W) (Figure 4b). It should be noted that these two correlation clusters interestingly represented the new and old amino acids, respectively. In general, the amino acids showed a positive correlation in usage within the clusters and a negative correlation between the clusters. Although the correlation analyses of the codon and amino acid usage exhibited similar patterns, no one-to-one correspondence was noted between the clusters. This indicates that amino acid usage may be independent of single codon usage. However, we identified consistent crosstalk between the amino acids and codons when we degenerated the codons by fixing their first two bases (Figure 4b). Similar analyses were also undertaken for changing the first or second base of the codons, but no consistent crosstalk between clusters of amino acids and codons was observed (Appendix A). This implies that amino acids, not codons, might be the dominant factor affecting usage. That is, life has introduced the mechanism of codon degeneracy to neutralise frequent genetic mutations in the primordial environment. However, the amino acids themselves may not be key factors in the selection of the ending base pair, whether G/C or A/T, for codons during the process.

## 4. Discussion

Thus far, valuable work effort has been expended to reveal the differential usage of genetic codons and amino acids on the whole-genome scale [15,32,33]. Du et al. ascribed energy effects to the impacts of GC content on the amino acid composition of proteins [16]. Gregory et al. linked the composition of genomic nucleic acids with the deviation of codon/amino acid usage by analysing nucleotide contents across 21 different bacterial and archaeal species [34]. Bharanidharan et al. identified a correlation between amino acid usage and GC content by analysing the use of a single nucleotide or dinucleotide in 115 bacterial species [35]. Other studies [36,37,38] have reported that the nucleotide composition of each taxon’s genomic DNA may affect codon selection and protein amino acid composition. However, many aspects have not yet been completely explained, such as the amino acid composition of the LUCA protein, the recruitment order of amino acids in early proteins, and so on. In this study, we acquired genomic and protein-coding information of 7270 species from public zones, covering sufficient representatives from the three domains of life. Using these pieces of information, we carried out a retrospective analysis of amino acid and codon usage on large scale, which enabled us to trace the recruitment order of amino acids in the early stages of the proteogenesis process and to determine the amino acid composition of the LUCA protein.

### 4.1. Amino Acid Usage Bias Is Ubiquitous and Is Independent of Single Codon Usage Bias

Theoretically, possessing more codons should grant amino acids more chances to be coded. However, many previous studies, as well as our results (Figure 1), indicate that the bias of amino acid usage is ubiquitous across diverse species. This bias is also bidirectional; some codon-rich amino acids are under-used and some codon-less amino acids are over-used. For example, the use of arginine (encoded by six codons) would be estimated at 9.8% [20], but it only reaches approximately 5.3% in most species. The use of lysine and glutamic acid, which possess only two codons each, is 5.3% and 6.2%, respectively. Indeed, amino acids take advantage of possessing more codons. In this study, amino acid usage showed a weak positive correlation with codon number, particularly in animals belonging to Eukaryota. However, such correlations were weaker in the representatives of Bacteria and Archaea. Therefore, codon number is not the sole factor to determine amino acid usage in proteins but may be a major factor.

What causes amino acid usage bias? This is a challenging question that has not yet been answered. Several previous studies ascribe amino acids usage bias to codon usage bias. Burge et al. analysed the relative abundance of dinucleotides in prokaryotes, eukaryotes, organelles and viruses and found that the CG dinucleotide was strictly used in many genomes [39]. Broader genome-wide analyses performed in this study also show that CGN usage is significantly lower than GCN usage. Osawa et al. linked codon usage bias with variation of protein evolution in early organisms [20]. Berezovsky et al. proposed that codon usage bias may be related to a significant reduction of the degree of freedom of the arginine side chain after protein composition [40]. Relative to arginine, the other alkaline amino acid, lysine, is encoded by only two codons but is over-used in proteins, at a rate of 5.3%. A reasonable explanation of the over-use of lysine is that it requires less energy for synthesis in both aerobic and anaerobic conditions than arginine [41] and it takes less enthalpy change during formation [42].

Codon usage bias has a large impact on amino acid usage bias. This phenomenon has been recognised and studied for a long time. Conventionally, codon usage bias is bound with the GC content in the species’ genome. Closely related taxa usually share patterns of codon usage and codon aversion; these patterns are phylogenetically conserved [43]. However, the multispecies analyses conducted in this study found that codon usage was discrepant in species with either the same or different GC content. In addition to GC content, several other factors may also affect codon usage, such as species self-selection [44], mutation bias, insertion bias [45], strand-specific nucleotide bias [46], CpG bias [47], GC/AT bias [34,48] and so on. Based on the usage profiles of over 7270 species, we identified two correlated clusters for codons and amino acids separately (Figure 4). Interestingly, no one-to-one correspondence was observed between the clusters of codons and amino acids, which indicates that amino acid usage may be independent of single codon usage. However, we identified consistent crosstalk between amino acids and codons when we degenerated codons by fixing their first two bases. Hence, codon degeneracy not only eliminates a large number of harmful mutations but also countermines the rapid but fatal changes in protein composition that allows for adaptation in complex living environments [16]. The cause-effect relationship between amino acid usage bias and codon usage bias is controversial. The two have likely been mutually influenced over the long course of evolution. The usage status in extant species is the consequence of a temporary balance of mutual influence between amino acids and codons.

### 4.2. Codon Usage Bias Is the Imprint of the Speciation Process

Yang et al. utilized a suite of codon models, including a Markov-based estimate of codon substitution rate to reconstruct ancestral genes with PAML4 [49]. Gil et al. used hundreds of codon models to evaluate phylogenetic branches with CodonPhyML [50]. In conjunction with other methods, codon usage information could serve as strong support for constructing species trees [51]. The phylogenetic trees of the codon and amino acid usage of species may reflect the relationship between species and genera in the three domains of life. Our data also indicated that the phylogenetic tree of codon usage is characterized by greater potential for exploration and clarification of phylogenetic relationships at an early stage of life evolution compared with amino acid usage. In conjunction with other methods, analysis of codon can provide support to proposed species trees [51]. In this study, we performed pseudotime analysis to estimate the quasi-evolutionary time of species emergence based on codon usage profiles. This constitutes a new attempt that has not been made previously. The quasi-evolutionary time of species consolidates previous findings that bacterial and archaeal species may pre-date eukaryotes [52]. Of the studied species, *green algae* are thought to have been the first eukaryotes, which is consistent with the earliest eukaryote fossil data on *Grypania spiralis* [30]. The present results are not completely consistent with those of some other studies, where for example Aquifex is suggested as the earliest bacteria that branched from Archaea [53]. Currently, most phylogenetic trees are constructed from molecular data such as homologous DNA or protein sequences, which only reflect the evolutionary relationship between species and cannot truly represent the evolutionary relationship of entire species. As described in our study the strategy of reconstructing the evolutionary relationships among species using the genomic codon usage profile proved to be a powerful tool in evolutionary studies. However, this strategy only introduces information on the coding region of species, omitting information on non-coding genome regions that also play an important role in life evolution [54]. The results of the pseudotime analysis support the hypothesis that Bacteria appeared first, followed by Archaea and then Eukaryota. This result agrees with previous hypotheses that eukaryotes evolved from archaeal species [52,55].

### 4.3. Recruitment of Amino Acids into Proteogenesis Is a Progressive Evolution Process

The available information on amino acid usage calls into question the assumption that the current 20 amino acids were recruited into the protein composition of early life forms simultaneously. The recruitment of all 20 amino acids in proteogenesis is too extravagant for early life to have survived in a primordial environment, which may require a complex biosystem and sufficient energy to support this process. In the early ecosystem, proteins composed of 10 amino acids may have been sufficient to drive the processes of life [31]. If the amino acids were not recruited simultaneously, then which ones were the first? Jordan et al. suggested that amino acids with decreased usage in proteins were likely to have been the first amino acids to have been recruited, and vice versa [12]. Du et al. [16] provided support for this view by analyzing the possible LUCA genes and protein in *Methanococcus maripaludis S2* [56]. Logically, biosynthesis can be expected to take advantage of energetically cheaper [57]; thus the amino acids encoded by codons with high GC content may have been preferred by LUCA [58]. Hence, amino acids encoded by codons with high GC-content may have been the first recruited into proteins [7,18]. This conception is supported by this study’s results, indicating that species with short quasi-evolution time tended to have high GC content. We confirmed that most early amino acids likely composing of LUCA’s protein were lost over the course of evolution [12,13,59]. Previous studies have also found that newly recruited amino acids require time to establish a foothold via sacrificing the usage of the original amino acids [7,11,12]. Recent advances in research on early amino acids found that they had optimal protein folding skeletons [60] and could construct a stable protein in vitro experiments [61]. By analyzing the gain and loss of amino acids over the three domains of life, we propose that 10 amino acids (A, D, E, G, L, P, R, S, T and V) may have been the first batch of early amino acids recruited into proteins in early life forms.

We previously analyzed amino acid usage in open reading frames of 549 taxa, proposing the chronological order of amino acids (L, A, V/E/G, S, I, K, T, R/D, P, N, F, Q, Y, M, H, W, C) based on amino acid usage. In addition, based on Pearson correlation coefficient and rank of amino acid usage, combined with the encoding amino acid codons, we previously proposed two conserved evolutionary paths of amino acids (A→G→R→P and K→Y) [15]. In this study, in addition to Miller’s experiments and the use of gain and loss, several chemicals, biological and geological factors were also considered to have aided the determination of the recruitment order of amino acids. For instance, considering that early proteins must have adapted to atmospheric changes, the amino acids that had ultraviolet absorption (C, F, H, W and Y) would be recruited later into proteins. In addition, H and W are heterocyclic aromatic amino acids, for which the compound complexity and the formation enthalpy [42] are relatively large and noneconomic for early life. Increased serine (S) use was considered to have had late absorption [7,62]. By integrating the multidimensional information of usage rank, usage correlation, chemical similarity and the upstream and downstream relationships of biological synthesis, we proposed two potential recruitment routes for amino acids, as follows: I→F→Y→C→M→W→U and K→N→Q→H→O.

It is noteworthy that the evolution of amino acids is still ongoing. The use of old amino acids is diminishing and new amino acids are emerging and are beginning to establish their foothold, replacing the older ones in the protein. For instance, selenocysteine (Sec, U), which is formed by the substitution of sulfur on cysteine with selenium, has been found to exist in several enzymes, such as glutathione peroxidase, thyroxine 5′-deiodinase, thioredoxin reductase, formate dehydrogenase, glycine reductase and some hydrogenases. Pyrrolysine (Pyl, O) is formed by inserting a group (4-methylpyrroline-5-carboxylate) in amide linkage with the ^ε^N of lysine. It anticipates protein biosynthesis in some methanogenic archaea and bacteria [63].

## 5. Conclusions

In conclusion, based on the results of genome-wide analyses of amino acid and codon usage, we propose that amino acid usage bias is a ubiquitous phenomenon and is independent of single codon usage bias. We also suggest a novel chronological order of the emergence of the three domains of life. Furthermore, we infer that the amino acids A, D, E, G, L, P, R, S, T and V might have first been recruited into LUCA protein, with the remaining amino acids being recruited into proteogenesis gradually along the two parallel routes of I→F→Y→C→M→W and K→N→Q→H across a long evolutionary period. This study provides new insight into the origin of life, illustrates the evolution of protein in early life and inspires a new landscape for evolutionary studies on extant species.

## Figures and Tables

**Figure 1 biomolecules-12-00171-f001:**
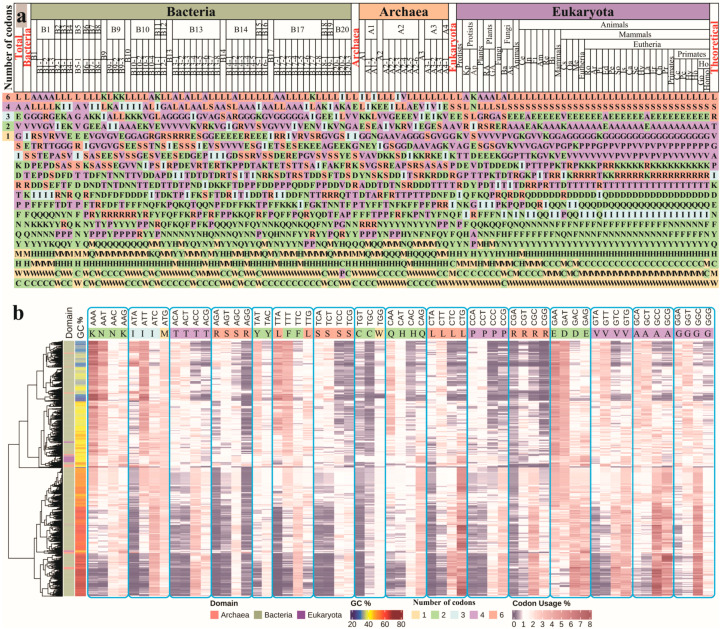
Amino acid and genetic codon usage landscapes in the three domains of life. (**a**) Results of the ranking of amino acids usage in species. The results are displayed from high to low, where different colours represent the different number of codons encoding amino acids. The columns headed Total, Bacteria, Archaea and Eukaryota indicate the order of amino acid usage by all species, all bacterial species, all archaeal species and all eukaryotes, respectively. The last column represents the theoretical ranking of amino acid usage. Other columns represent the amino acid usage ranking of species (see Appendix A for abbreviations). (**b**) Heatmap analysis of codon usage (excluding stop codons, ComplexHeatmap R package). Hierarchical clustering of the GC content and domain of life of each species (*n* = 7270). The horizontal bars indicate the codon and amino acid.

**Figure 2 biomolecules-12-00171-f002:**
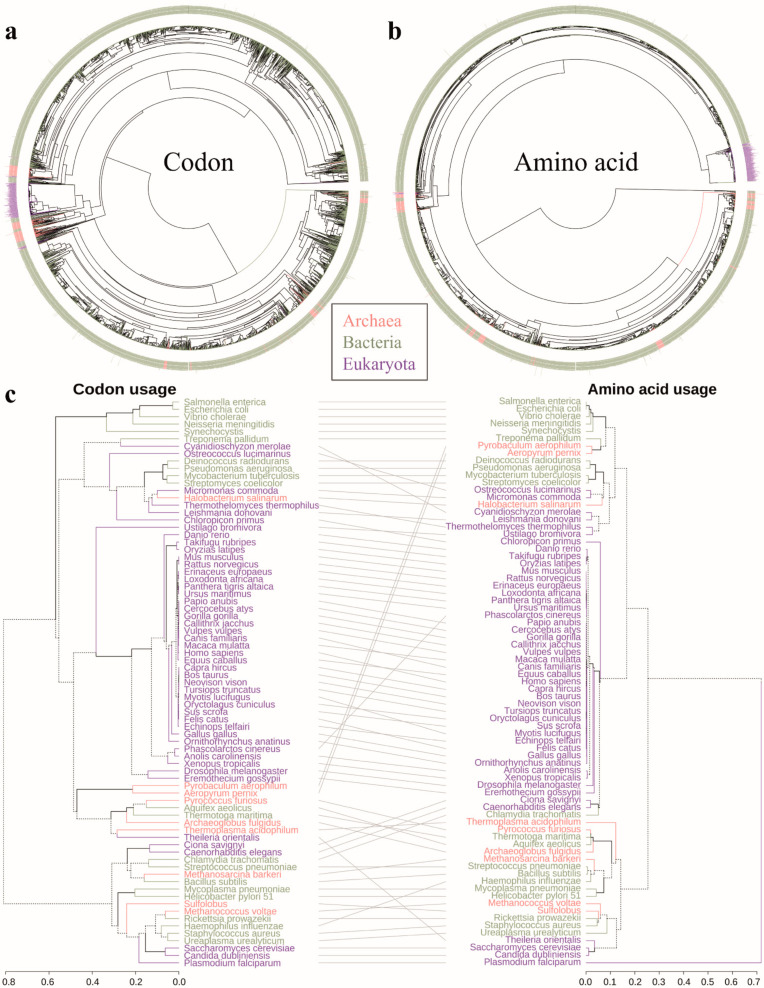
Phylogenetic analysis of the included species. (**a**) Phylogenetic tree of the codon usage profiles of the species. (**b**) Phylogenetic tree of the amino acid usage profiles of the species. (**c**) Comparison of phylogenetic trees of codon and amino acid usage in model organisms (Appendix A). Note: Pink, green.; purple represent Archaea, Bacteria and Eukaryota, respectively. These figures were constructed using the dendextend and pvclust R packages.

**Figure 3 biomolecules-12-00171-f003:**
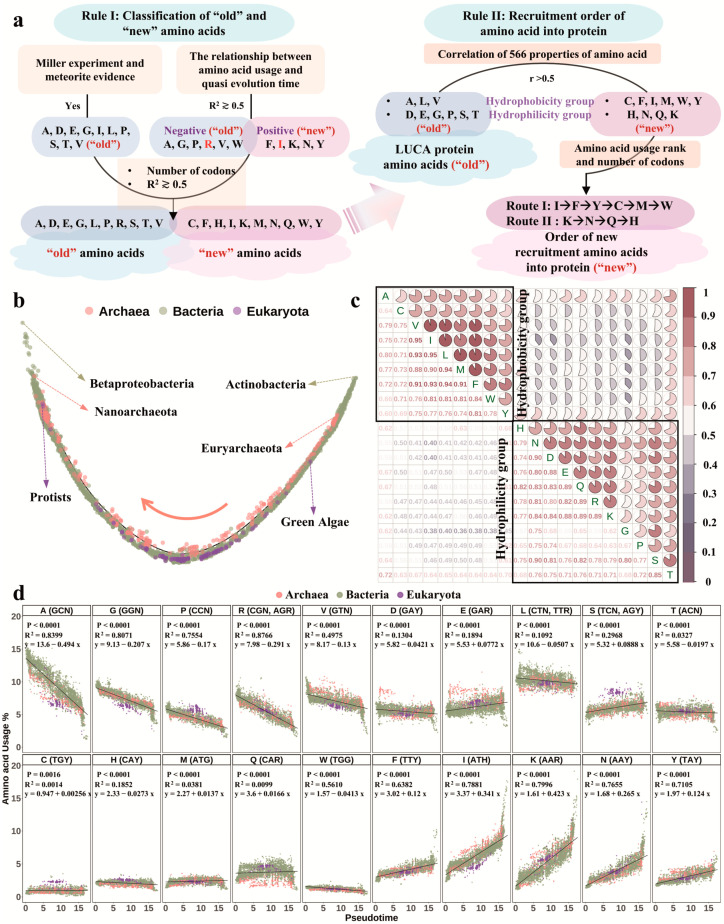
Amino acid composition of LUCA protein and order of recruitment of the remaining amino acids into the protein. (**a**) Overview of the classification and recruitment of new and old amino acids in proteogenesis. (**b**) Pseudotime analysis of species (monocle 2.20.0 R package). The results show the chronological order of the three domains of life as follows: Bacteria→Archaea→Eukaryota. Each dot represents a species, the colours represent different domains of life, and the arrow indicates the evolutionary direction of the species. (**c**) Correlation of 566 physicochemical properties of 20 amino acids (corrplot R package). The top right triangle represents the correlations between the physicochemical properties of amino acids. The bottom left triangle indicates the correlation value between amino acids, where red indicates high correlation (r > 0.5), and blue indicates low correlation. (**d**) Correlation of amino acid usage and quasi-evolution time (ggplot2 R package). The abscissa represents quasi-evolution time, and the ordinate represents amino acid usage (%). The codons are given in parentheses in the figure. Note: N represents any bases; R represents A and G bases; Y represents T and C bases; and H represents T, C and A bases.

**Figure 4 biomolecules-12-00171-f004:**
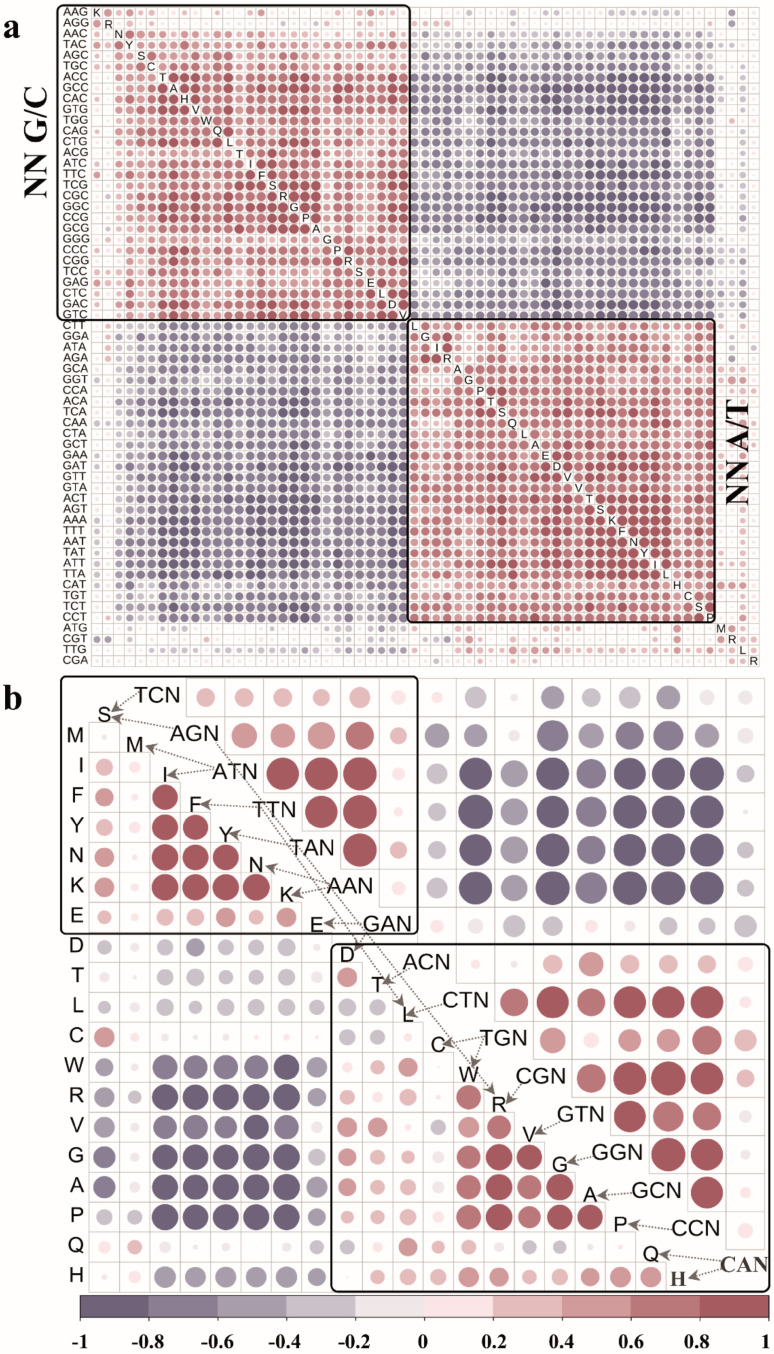
Correlation analysis of amino acids/genetic codons. (**a**) Correlation analysis of codon usage (excluding stop codons) in all species. Note: N represents any base. (**b**) Correlation analysis of amino acid usage (**bottom left**) and codon usage with the same first two bases (**top right**) in all species. Note: Single-letter abbreviations denote amino acids. Arrows indicate the correspondence between codons and amino acids. The figures were constructed using the corrplot R package.

**Table 1 biomolecules-12-00171-t001:** Fold change of codon usage by species groups with different GC contents.

GC%	<45	≥45	FC	GC%	<45	≥45	FC
avgGC%	36.60	59.10	1.61	avgGC%	36.60	59.10	1.61
**AAA (K)**	5.54 ± 1.93	1.60 ± 1.29	0.29	**AAT (N)**	3.70 ± 1.44	1.08 ± 0.68	0.29
**AAG (K)**	1.94 ± 0.81	2.06 ± 0.73	1.06	**AAC (N)**	1.57 ± 0.50	1.94 ± 0.33	1.23
**ATA (I)**	2.04 ± 1.61	0.42 ± 0.58	0.21	**ATT (I)**	4.19 ± 1.23	1.51 ± 1.11	0.36
**ATG (M)**	2.33 ± 0.32	2.33 ± 0.39	1.00	**ATC (I)**	1.68 ± 0.67	3.21 ± 0.74	1.91
**TTA (L)**	3.62 ± 1.60	0.55 ± 0.67	0.15	**TTT (F)**	3.45 ± 0.94	1.27 ± 0.90	0.37
**TTG (L)**	1.66 ± 0.64	1.27 ± 0.73	0.76	**TTC (F)**	1.20 ± 0.48	2.32 ± 0.64	1.94
**TAA (Stop)**	0.21 ± 0.07	0.10 ± 0.07	0.51	**TAT (Y)**	2.61 ± 0.70	1.10 ± 0.60	0.42
**TAG (Stop)**	0.06 ± 0.03	0.06 ± 0.03	0.90	**TAC (Y)**	1.05 ± 0.38	1.49 ± 0.43	1.42
**ACA (T)**	1.87 ± 0.53	0.55 ± 0.40	0.30	**ACT (T)**	1.68 ± 0.45	0.53 ± 0.37	0.31
**ACG (T)**	0.75 ± 0.42	1.70 ± 0.61	2.27	**ACC (T)**	1.00 ± 0.56	2.67 ± 0.74	2.66
**AGA (R)**	1.25 ± 0.65	0.26 ± 0.30	0.21	**AGT (S)**	1.43 ± 0.34	0.50 ± 0.33	0.35
**AGG (R)**	0.44 ± 0.44	0.37 ± 0.51	0.83	**AGC (S)**	0.97 ± 0.52	1.63 ± 0.38	1.68
**TCA (S)**	1.47 ± 0.46	0.47 ± 0.33	0.32	**TCT (S)**	1.62 ± 0.56	0.48 ± 0.37	0.30
**TCG (S)**	0.47 ± 0.24	1.41 ± 0.56	2.98	**TCC (S)**	0.62 ± 0.46	1.23 ± 0.44	1.99
**TGA (Stop)**	0.12 ± 0.22	0.17 ± 0.08	1.44	**TGT (C)**	0.64 ± 0.25	0.27 ± 0.18	0.43
**TGG (W)**	0.93 ± 0.28	1.40 ± 0.17	1.51	**TGC (C)**	0.36 ± 0.28	0.68 ± 0.19	1.88
**CAA (Q)**	2.57 ± 0.87	0.99 ± 0.70	0.39	**CAT (H)**	1.32 ± 0.31	0.90 ± 0.40	0.69
**CAG (Q)**	1.19 ± 0.83	2.71 ± 0.60	2.28	**CAC (H)**	0.63 ± 0.33	1.28 ± 0.39	2.03
**CTA (L)**	1.01 ± 0.38	0.34 ± 0.33	0.34	**CTT (L)**	1.86 ± 0.62	0.98 ± 0.62	0.53
**CTG (L)**	1.01 ± 1.03	4.95 ± 1.71	4.91	**CTC (L)**	0.72 ± 0.52	2.29 ± 1.28	3.19
**GAA (E)**	4.80 ± 0.97	2.70 ± 1.12	0.56	**GAT (D)**	3.88 ± 0.70	2.25 ± 0.95	0.58
**GAG (E)**	1.84 ± 0.91	3.12 ± 1.18	1.69	**GAC (D)**	1.36 ± 0.61	3.31 ± 1.26	2.44
**CCA (P)**	1.29 ± 0.39	0.53 ± 0.35	0.41	**CCT (P)**	1.29 ± 0.35	0.57 ± 0.32	0.44
**CCG (P)**	0.54 ± 0.38	2.46 ± 0.78	4.55	**CCC (P)**	0.47 ± 0.46	1.40 ± 0.66	2.94
**CGA (R)**	0.44 ± 0.24	0.41 ± 0.23	0.94	**CGT (R)**	1.02 ± 0.60	1.10 ± 0.53	1.08
**CGG (R)**	0.29 ± 0.30	1.39 ± 0.81	4.86	**CGC (R)**	0.57 ± 0.40	3.01 ± 1.15	5.26
**GCA (A)**	2.29 ± 0.57	1.36 ± 0.64	0.60	**GCT (A)**	2.39 ± 0.58	1.17 ± 0.63	0.49
**GCG (A)**	0.95 ± 0.60	3.95 ± 1.61	4.15	**GCC (A)**	1.16 ± 0.70	4.65 ± 1.75	4.00
**GGA (G)**	1.97 ± 0.75	0.88 ± 0.49	0.45	**GGT (G)**	2.19 ± 0.67	1.46 ± 0.61	0.67
**GGG (G)**	0.87 ± 0.41	1.39 ± 0.52	1.61	**GGC (G)**	1.22 ± 0.63	4.34 ± 1.35	3.56
**GTA (V)**	1.80 ± 0.64	0.67 ± 0.47	0.37	**GTT (V)**	2.42 ± 0.64	1.07 ± 0.69	0.44
**GTG (V)**	1.30 ± 0.69	3.11 ± 0.85	2.39	**GTC (V)**	0.86 ± 0.48	2.67 ± 1.15	3.11

Codon usage is represented by average codon usage (%) ± standard deviation of all species within the corresponding threshold. Amino acid abbreviations are shown in parentheses. The calculation method of fold change (FC) value is given in Section 2. avgGC%: average GC content.

## Data Availability

The authors declare that all data supplementary to the findings of this study are available within the paper and its Appendix A or from the corresponding author upon reasonable request.

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
