# Peer review of "Determination of the Amino Acid Recruitment Order in Early Life by Genome-Wide Analysis of Amino Acid Usage Bias"

_biomolecules, 2022, doi:10.3390/biom12020171_

Round 1

Reviewer 1 Report

see attached file for comments.

Reviewer 2 Report

Dear Editors,

Dear Authors,

The reviewed study entitled: “Genome-wide Analysis of Amino Acid Usage Bias to Unveil Amino Acid Recruitment Order in Early Life” represents valuable and interesting insight to the genomic and evolutionary studies. The reviewed study carries out the original analysis of genomic data available in NCBI data base. The used amount of data for more than 7k extant species and the extend of performed analysis is impressive. The study subject the issue of amino acids protein recruitment in proteins in the early history of life. The obtained results shed light not only on the genome structure differences and evolutionary relationships between three main domains of life, i.e., Arachea, Prokaryota and Eukaryot but also on functional understanding of life origin and evolutionary history of proteogenesis apparatus, particularly in terms of the basis protein composition and its change in time. The reviewed study characterizes by good quality and the employed measurements against specified aims are adequate. Language presentation is quite well understandable but require improvement - some significant corrections have been done by myself. Moreover, the Authors need to answer the questions placed in the attached file.

Unfortunately, the study is deficient in detailed description of applied methodology. In my opinion, due to high complexity and richness of available bioinformatic tools for genomic data processing it is very important and even crucial to describe, with details, the used methods of data collection and processing. Without such information no one can make any replicable analysis or use the data in the future. Therefore, I believe that every contemporary genomic-based bioinformatics should be provided with detailed descriptions of used software, including the data input type and format, analysis methods, settings, generated input/output data and procedure of the final results presentation. As a reviewer, I had troubles in evaluation the methodological part of the manuscript without detailed information. In the manuscript there in no information how the data were selected form databases, how filtered, how stored, how processed in terms of codon identification, counting its number, graphics generation, etc.

In conclusion, I highly recommend to publish the reviewed manuscript in the Biomolecules Periodical, but Authors should complete their manuscript with information essential for the full manuscript evaluation and for future potential readers that would like to perform the data analysis in the manner described by Authors.

All fixes, questions and remarks were placed in the enclosed pdf file.

Thank you for another interesting manuscript that I could review!

Round 2

Reviewer 2 Report

Dear Editors,
Dear Authors,

The revised manuscript has been essentially improved and in my opinion is ready for publication in the Biomolecules. The reviewed study characterizes by very good quality and the employed measurements against specified aims are rich and adequate. Language presentation is very well.

It is a very good job! Congratulations!